# Rye Dietary Fiber Components upon the Influence of Fermentation Inoculated with Probiotic Microorganisms

**DOI:** 10.3390/molecules28041910

**Published:** 2023-02-16

**Authors:** Kamila Koj, Ewa Pejcz

**Affiliations:** Department of Fermentation and Cereals Technology, Wrocław University of Environmental and Life Sciences, 51-630 Wrocław, Poland

**Keywords:** dietary fiber, rye bread, sourdough fermentation, probiotics, arabinoxylans, fructans, β-glucans

## Abstract

Rye flour is used as the main ingredient of sourdough bread, which has technological and gastronomic benefits and increased nutritional value. The transformations observed during fermentation and baking may enable the conversion or degradation of rye dietary fiber carbohydrates built mainly of arabinoxylans, fructans, and β-glucans. This study aimed to determine the dynamics of the changes in the contents of complex carbohydrates in sourdoughs inoculated with potential probiotic microorganisms as well as the polysaccharide composition of the resulting bread. Sourdoughs were inoculated with the potential probiotic microorganisms *Saccharomyces boulardii*, *Lactiplantibacillus plantarum*, *Lacticaseibacillus rhamnosus*, and *Bacillus coagulans*, and spontaneous fermentation was performed as a control. Samples of the sourdoughs after 24 and 48 h of fermentation and of bread obtained with these sourdoughs were analyzed for the content of individual dietary fiber components. The present study demonstrated that the treatments applied contributed to an increased total content of arabinoxylans in the breads, and the inoculation of the sourdoughs with the potential probiotic strains improved their solubility in water. The use of the *S.boulardii* strain may seem prospective as it allowed for the greatest reduction in fructans in the rye bread. Rye sourdough bread is an attractive source of dietary fiber and can be modified for different nutritional needs.

## 1. Introduction

Rye is the second most commonly used grain for bread making after wheat cereal. The main constituents of its grain include starch (57.1–65.6%), dietary fiber (14.7–20.9%), protein (9.0–15.4%), and ash (1.8–2.2%) [1]. Dietary fiber is built of carbohydrate polymers containing three or more monomeric units, which cannot be absorbed in the small intestine, and is commonly classified based on its solubility in water. Soluble dietary fiber is fermented in the colon, and this process results in the formation of short-chain products, including lactate, butyrate, acetate, and propionate. This dietary fiber fraction reduces the serum cholesterol level and the postprandial blood glucose level [2]. It is also effective in mitigating the general symptoms of irritable bowel syndrome (IBS) [3]. In turn, the insoluble dietary fiber fraction is effective in softening fecal bulk and shortening the time of stool passage through the colon [4]. This fraction of rye dietary fiber is mainly built of arabinoxylans, fructans, and β-glucans with the first mentioned compounds being the most abundant in rye among all cereal grains. The content of these carbohydrates in rye grain affects the water-binding capacity and viscosity of dough [5]. Arabinoxylans are dimers of arabinose and xylose [1]. They are divided into water-soluble and water-insoluble fractions depending on their molecular weight, substitution pattern (arabinose to xylose ratio), and cross-linking degree [6]. Their content in the rye grain ranges from 7% to 12% [7]. Arabinoxylans are proven to elicit great benefits to human health by their immunomodulatory activity and their impact on reducing cholesterol levels, as well as by alleviating symptoms of type 2 diabetes, increasing absorption of certain minerals, and causing the stool-swelling effect [8]. Given the above-mentioned effects, the water-soluble arabinoxylans of rye positively affect enteral fermentation and the production of short-chain fatty acids [9]. In turn, fructans are carbohydrates built from fructose monomers of various polymerization degrees, and some of them possess a terminal D-glucose unit. In general, they are classified based on the differences in the polymerization degree and bonds between monomers. Their content in the rye grain ranges from 4.3% to 5.0% [9]. Fructans increase the sensation of satiety, thereby contributing to lower energy consumption. They also enhance calcium absorption and decrease the serum concentration of triglycerides [2]. β-Glucans are homopolysaccharides built from glucose monomers linked by means of β-(1→3), β-(1→6), β-(2→3), or β-(3→6) glycosidic bonds. They are classified as the water-soluble fraction of dietary fiber, although they also contain some water-insoluble fractions [2]. Their content in the rye grain ranges from 2% to 3% [7]. They are capable of producing highly viscous solutions, thereby exerting positive physiological effects [10].

Fermentable oligosaccharides, disaccharides, monosaccharides, and polyols (called FODMAPs), which include the fermentable carbohydrates of the rye grain, may elicit both positive and negative impacts on human health. On the one hand, they may serve as a medium for intestinal microbiota, but on the other hand, they may trigger unbeneficial symptoms associated with IBS. The prebiotic effect of oligosaccharides relates to bacterial inversion resulting in the production of short-chain fatty acids, which improve the properties of the intestinal barrier and stimulate resistance to various intestinal infections [7]. The FODMAPs are dietetic triggers of symptoms of the functional disorders of the intestines, owing to which it is feasible to treat these symptoms [11]. A low-FODMAP diet may confer beneficial effects by mitigating IBS symptoms but may also pose a risk associated with insufficient dietary fiber intake and malnutrition. Thus, it is essential to provide the body with the appropriate amounts of dietary fiber and nutrients, which are indispensable for its proper development and functioning [12].

Bread is one of the most important staple foods consumed by people around the world, and by this means, the main source of the intake of whole-grain cereals. The emergence of industrial baking has made yeast fermentation superior over sourdough fermentation. Recently, however, sourdough and the bakery products made with its use have regained their popularity, not only due to technological and gastronomic benefits, but also due to their increased nutritional value and health benefits [13]. Rye flour is commonly used as the main ingredient of sourdough bread [1]. Traditional sourdough is produced as a result of the spontaneous fermentation of flour and water with homo- and hetero-fermentative lactic acid bacteria (LAB) and yeast. So far, approximately 50 LAB strains have been isolated from sourdough, including *Enterococcus*, *Lactococcus*, *Leuconostoc*, *Pediococcus*, *Streptococcus*, and *Weissella*, but the major isolates have been the strains of *Lactiplantibacillus*, including *Lb. Brevis*, *Lb. Sanfranciscensis*, and *Lactiplantibacillus plantarum*. Sourdough also contains over 20 species of yeast, the most popular of which is *Saccharomyces cerevisiae.* Undoubtedly, the vast differences in the identified yeast number and species depend on such factors as the type of cereal grain used, dough hydration process, leaven temperature, and fermentation conditions [14].

The transformations observed during fermentation and baking may enable the conversion or degradation of carbohydrates without reducing the total dietary fiber content of the bread. Sourdough bread making is generally based on the capabilities of LAB representing a reservoir of extracellular fructanases or glucanases, which enhance the metabolic potency of the fermentative microbiota converting saccharose into non-digestible polysaccharides or oligosaccharides. An extended fermentation time intensifies the involvement of flour enzymes in the modifications and degradation of the dough ingredients [7,15]. Sourdough fermentation is proven to trigger the conversions of lipids, macromolecules, and several phytochemicals, including folates, phenolic acids, and bioactive compounds [13]. It has also been shown to affect the breads’ texture and taste due to the formation of flavor compounds and also its nutritional value [10].

In recent years, allochthonous sourdough starters containing, i.a., probiotic microorganisms have been used to control fermentation processes [14]. According to the 2002 report issued by the World Health Organization (WHO), probiotics are live microorganisms that have a positive effect on host health when administered in appropriate amounts. Most often, however, probiotic preparations are used in clinical practice, including bacteria of the *Lactiplantibacillus*, *Bifidobacterium*, and *Streptococcus* genera, as well as *Saccharomyces* yeast [16]. The consumption of functional food with therapeutic and health-promoting effects by consumers imposes new requirements as to its quality [17]. The fact that most probiotics are inactivated during the baking process poses a serious problem in this case. However, recent studies show that dead or inactivated probiotics retain their beneficial effects on the host due to their immunomodulatory activity [18]. *Saccharomyces boulardii* is known as a potential probiotic yeast. It is a variety of *Saccharomyces cerevisiae*, which can be distinguished from other strains of this genus based on diversified molecular criteria. Even though it is a valuable microorganism, it is rarely used in the food industry; however, it has been deployed for the fermentation of cereals and their derivative products [19]. The strain *Lpb. plantarum* is highly tolerant to acids; hence, it is often the major strain involved in the fermentation of cereals, especially considering its capability for transporting and metabolizing various plant carbohydrates [20]. In turn, *Lacticaseibacillus rhamnosus* is a facultative heterofermentative bacterium fermenting hexoses (e.g., fructose) to lactic acid and also pentoses to a mixture of lactic and acetic acids [21]. When it comes to the strain *Bacillus coagulans*, it is generally recognized as safe under the conditions of its intended use [22]. It may survive in an environment of high temperature, acidity, and salinity; as a result, it may be claimed a real candidate for use in high-temperature-treated products [23].

There are few reports so far on the benefits of using inoculums of potential probiotic microorganisms for the production of bread with a modified composition of complex carbohydrates. Previous research focused on isolating various microorganisms from spontaneously fermenting sourdoughs and not on their effect on the metabolism of the individual flour components during fermentation. The use of a single-component inoculum for fermentation perfectly indicates the direction of changes in the carbohydrate content occurring when using a particular potential probiotic strain. The production of bread with the use of those strains can increase the diversity of the cereal product range, leading to a satisfaction of the consumers’ needs. Knowledge of the metabolism of these strains gives great prospects for the future in order to create bread with a specific carbohydrate composition, e.g., with reduced FODMAP content for people with IBS. This study aimed to determine the dynamics of the changes in the contents of complex carbohydrates in sourdoughs prepared with potential probiotic microorganisms. Its second aim was to determine the polysaccharide composition of the sourdough bread made with these microorganisms.

## 2. Results and Discussion

### 2.1. Rye Flour Composition

Table 1 presents the contents of individual polysaccharides in the rye flour and the bread’s dry matter. Starch accounted for 58.67% of the flour composition. The content of total arabinoxylans reached 8.12%, including 2.11% of its soluble fraction and 6.01% of its insoluble fraction. Fructans accounted for 3.82% of the flour composition, whereas β-glucans accounted for 1.07% of the flour composition. In turn, the dietary fiber content reached 10.39%, including 1.12% of its soluble fraction and 9.27% of its insoluble fraction. The rye flour used in this study had appropriate quality parameters for bread baking. The chemical composition of rye flour depends mainly on genetic factors, soil quality, and climatic and cultivation conditions [1].

### 2.2. Contents of Individual Complex Carbohydrates in the Sourdoughs

The metabolism of carbohydrates during sourdough fermentation depends, to a various extent, on the microbiological composition of the starter; availability of substrates, NaCl, and microbiological enzymes derived from the flour; and such process parameters as dough yield and fermentation time and temperature [24,25]. Fermentation is based on the hierarchy of the consumption of various sugars with glucose consumed in the first place [26].

The present study showed a positive effect of using the potential probiotic strains to increase the content of soluble arabinoxylans in all the sourdoughs. Figure 1a–e depicts the changes in the contents of the polysaccharides (dry matter) during the fermentation of the sourdoughs inoculated with strains of various microorganisms. In the case of the spontaneously fermenting sourdough (Figure 1a), the highest content of total arabinoxylans and insoluble arabinoxylans was determined after 24 h of fermentation, which was further observed to decrease but not to the baseline level observed before fermentation. The content of soluble arabinoxylans decreased in this sourdough after 48 h of fermentation compared to its level before fermentation, whereas the solubility decreased from 36% to 35%. In the sourdough inoculated with the *S.boulardii* strain (Figure 1b), the content of total arabinoxylans was observed to increase along with the fermentation time, whereas the content of the insoluble fraction was the lowest after 24 h of fermentation (4.52%) and the highest after 48 h of fermentation (5.21%). Under the influence of fermentation, the solubility increased by 16%. In the sourdough inoculated with the *Lpb.plantarum* strain, the highest contents of total and insoluble arabinoxylans were determined after 24 h of fermentation; however, with fermentation proceeding, the content of total arabinoxylans decreased to the initial value, and that of the insoluble fraction remained stable (Figure 1c). The use of this strain resulted in as much as a 18% higher solubility of arabinoxylans after 48 h of fermentation. Arabinoxylan is known to be degraded by cereal enzymes during dough resting, which contributes to its solubilization [27]. In the case of the sourdough inoculated with the *L.rhamnosus* strain (Figure 1d), the highest content of total arabinoxylans was determined after 24 h of fermentation (7.76%); it was observed to decrease with fermentation time but not to the level before fermentation. In turn, the content of insoluble arabinoxylans decreased over the entire fermentation period. The solubility of the arabinoxylans increased in this sourdough by 15% during fermentation. The use of the *B.coagulans* strain achieved the highest contents of total and insoluble arabinoxylans after 48 h of fermentation of the sourdough (Figure 1e). Among all sourdoughs inoculated with the potential probiotic strains, the latter mentioned strain caused the smallest increase in arabinoxylan solubility, i.e., by barely 4%. Acidic conditions positively affect the swelling of arabinoxylans, simultaneously suppressing the activity of xylanases and, thereby, reducing the degradation of arabinoxylans [28]. Given that soluble arabinoxylans feature high capabilities for water binding, gas retention in dough, and retarding stalling rate, they are highly suitable and desirable for rye bread making [6].

Rye represents a rich source of fructans, which increase the abundance of bifidobacteria in the gastrointestinal tract and enhance the absorption of metals, thereby diminishing appetite [2,9]. It has been shown that even a simple technological treatment decreases the content of fructans in rye bread by approximately 3% [7]. In the case of the spontaneously fermenting sourdough and the sourdough inoculated with the *B.coagulans* strain, the content of fructans decreased significantly after 24 h, while it remained stable in the second analytical term (Figure 1a–e). The use of the *S.boulardii* strain caused the fructan content to decrease after 48 h of fermentation to 0.35%. The sourdoughs inoculated with the *Lpb. plantarum* and *L. rhamnosus* strains contributed to a successive decrease in the content of fructans along with the fermentation time (Figure 1c,d). Previous studies proved that the reduced content of FODMAPs, including mainly fructans, could be ascribed to the metabolic activity of the LAB strain, including i.a. *Lpb.plantarum* present in sourdough, and extended dough fermentation time, which contributes to the increased tolerance of these compounds by patients suffering from IBS [25,29,30]. LAB produce lactic and acetic acids, which decrease the dough’s pH, thereby allowing for the activation of a few specific enzymes capable of reducing FODMAP content [31]. Various FODMAPs derived from rye flour are fermented by intestinal microbiota, which triggers undesirable symptoms when its level exceeds ca. 0.3 g/kg body weight, namely ca. 15 g/day. Sourdough fermentation with the LAB strains featuring activity of fructanases enables the production of low-fructan bread [7]. The content of β-glucans decreased significantly after 24 h of fermentation and afterwards remained stable in the case of the spontaneously fermenting sourdough and the sourdough inoculated with the *Lpb.plantarum* strain (Figure 1a,c). A previous study showed that β-glucans might be a prebiotic released by LAB during sourdough fermentation, thereby increasing the viability of the probiotic (*Lpb.plantarum*) [32]. The use of *S.boulardii*, *L.rhamnosus*, and *B.coagulans* to prepare sourdoughs resulted in a decreased content of β-glucans along with the fermentation time (Figure 1b,d,e). The use of spontaneous fermentation and the *Lpb. plantarum* inoculum resulted in maintaining the content of β -glucans at higher levels during sourdough production.

The metabolism of carbohydrates differs depending on the LAB species and strain, sugar type, co-presence of yeast, and processing conditions. Vast changes occurred in the carbohydrate fractions during sourdough fermentation due to both the enzymatic activity of the flour and the metabolic transformations triggered by the enzymes secreted by the microorganisms [24]. LAB activity in sourdoughs decreases the contents of non-digestible oligosaccharides, fructans, and raffinose (belonging to FODMAPS) in flour [32].

### 2.3. Contents of Individual Complex Carbohydrates and Dietary Fiber in the Sourdough Breads

The fermentation of flour and water activates the endogenous enzymes of the flour, thereby triggering starch degradation to amylase and the release of maltodextrin and maltose [33]. In the present study, the highest content of starch was determined in the breads made of the sourdoughs inoculated with *S.boulardii* (58.79%) and *Lpb.plantarum* (58.27%). In contrast, the lowest content of starch was found in the bread made of the sourdough prepared with B.coagulans (55.91%). The fermentation of sourdough leads to an increase in its acidity and a decrease in its pH [25,29,31]. Lower dough pH facilitates the formation of resistant starch, which diminishes the starch digestibility and, consequently, blood glucose level [34]. However, high dough acidification during fermentation results in suppressed enzymatic activity in the first stages of baking and, by this means, in reduced starch degradation [28].

When comparing the contents of total arabinoxylans in the rye flour (8.12%) and breads (8.61–10.54%), it was found that they increased during bread making regardless of the production method deployed. The highest contents of total and water-insoluble arabinoxylans were determined in the bread made of the sourdough inoculated with the *B.coagulans* strain. In turn, the lowest content of total arabinoxylans was determined in the control bread and in the breads made of spontaneously fermenting sourdough and the sourdough inoculated with *L.rhamnosus*. The control bread had a lower content of soluble arabinoxylans compared to the sourdough breads, and the highest content of this fraction of arabinoxylans was found in the breads made of the spontaneously fermenting sourdough and the sourdough inoculated with the *Lpb.plantarum* strain. Soluble arabinoxylans serve the role of soluble dietary fiber, which contributes to proper body functions and prevents such physiological disorders as diabetes, obesity, hypercholesterolemia, neoplasms, or alimentary tract dysfunctions [35].

Fructans represent the major group of compounds belonging to the FODMAPs. A long fermentation time of bread dough may contribute to a 90% reduction in fructans and 74% reduction in total FODMAPs [36]. At the early stage of fermentation, all FODMAPs are almost completely degraded (except for polyols); however, saccharose, fructose, and glucose are completely degraded in the first stage of fermentation and at the end of baking [37]. The content of fructans determined in the analyzed breads (0.62–1.46%) was lower than in the flour (3.82%). The fructan content was identical in all samples, except for the bread made of the sourdough inoculated with the *S. boulardii* strain, in which it was significantly reduced. Investigations conducted by other authors demonstrated that sourdough-based rye breads had higher contents of fructans than rye breads made with yeast [31,36,38]. This difference is probably due to the activity of bacterial hydrolytic enzymes; however, it is also likely that endogenous enzymes are activated in the sourdough at lower pHs [38]. In spite of the fact that rye bread is the richest source of fructans (1.94 g/100 g), extending fermentation time may be a means to prevent the degradation of the fructans by invertase secreted by *S.cerevisiae* [9,39].

β-glucan serves various roles in the human body, including mainly activation of the immune system, reduction of the postprandial blood glucose level, and general improvement in glycemia and the sensitivity to insulin in persons with normoglycemia [2,37]. The lowest β-glucan content was determined in the control bread (0.30%). The use of sourdough, regardless of its type, contributed to greater retention of β-glucans in the breads. Among all the breads analyzed, the highest β-glucan content was determined in the bread made of the sourdough inoculated with the *L.rhamnosus* strain (Table 1). LAB can produce glucans, which are synthesized extracellularly by membrane-bound glycosyltransferase [16,40].

Dietary fiber has been shown to improve host metabolism by, i.a., the positive impact of short-chain fatty acids produced from fermentable dietary fiber [41]. Rye bread consumption increases the intake of cereal-derived dietary fiber, which is implicated in reducing the risk of the development of, i.a., colorectal cancer [30]. The technological process of rye bread making, including enzymatic and heat treatment, contributes to changes in the content and solubility of the dietary fiber components [42].

The analyses conducted in the present study show that the highest total dietary fiber content was determined in the bread made of the sourdough inoculated with the *Lpb. plantarum* strain (12.16%), and the lowest was in the bread made of spontaneously fermenting sourdough and the sourdoughs inoculated with *S.boulardii* and *B. coagulans* strains (Table 1). The increase in dietary fiber content in the sourdough-based bread is likely due to the production of exopolysaccharides (EPS) by the LAB and to resistant starch formation during baking [42]. EPS have recently spurred great interest due to their hydrocolloidal nature, enabling the manufacture of novel products that prevent chronic non-communicable diseases, such as irritable bowel syndrome, high cholesterol level, cardiopathies, colitis, ulcers, and cancers, and at the same time exhibiting immunomodulatory effects [34,40]. In the case of the soluble dietary fiber fraction, its highest content was determined in the bread made of the sourdough inoculated with *Lpb.plantarum* (2.21%) and in the control bread (1.66%); in contrast, its lowest content was found in the bread made of the spontaneously fermenting sourdough and in the breads made of the sourdoughs inoculated with the *S.boulardii*, *L.rhamnosus*, and *B.coagulans* strains (Table 1). The highest content of the insoluble dietary fiber fraction was determined in the breads made of the sourdoughs inoculated with the *Lpb.plantarum* and *L.rhamnosus* strains; in contrast, the lowest content was found in the other breads analyzed. The enzymatic activity of LAB may contribute to the increased solubility of dietary fiber [17]. The improved properties of the dietary fiber of sourdough-based bread contribute to its reduced glycemic index [34]. Conventional sourdough baking reduces and converts carbohydrate compounds in the rye flour; however, the degree of reduction and transformation depends on the fermenting organisms. Thanks to the combination of knowledge about the production technology of baked goods from rye flour with the knowledge of the metabolism of selected potential probiotic microorganisms, it is possible to design and create innovative food products that are currently not available on the market.

## 3. Materials and Methods

### 3.1. Materials

One type of rye wholemeal rye flour (Mona, Poland) was used for the sourdough preparation and bread making. Encapsulated potential probiotic microorganisms of the following monocultures were used in the study:-*S.boulardii* (CNCM I-745) (Biocodex, Warszawa, Poland)-*Lpb.plantarum* (DSM 9843) (Sanprobi SP.Z O.O.SP.K., Szczecin, Poland)-*L.rhamnosus* (DSM 14870) (Bayer Pharma, Warszawa, Poland)-*B.coagulans* (MTCC 5856) (Singularis Herbs Corporation Ltd., Lewes, DE, USA).

### 3.2. Sample Preparation

The study was divided into three stages. Stage I involved the preparation of rye sourdoughs with 200% extraction yield with the addition of starter cultures that were fermented for 24 and 48 h at a temperature of 30 °C and an air humidity of 85% in a fermentation chamber (IBIS, Szubin, Poland). The microorganism preparations were added directly to the sourdoughs in a dose of 10^9^ CFU/kg of flour. Spontaneously fermenting sourdough without inoculum served as a control sample. Samples of the sourdough were analyzed immediately after preparation as well as 24 h and 48 h after fermentation. In stage II, bread doughs were prepared in a Brabender’s farinograph (Duisburg, Germany) from the sourdoughs fermented for 48 h, the remaining amount of flour (75%), water (90%), and salt (1.5%). The control bread was prepared without the sourdough but with the addition of lactic acid (1%). Afterwards, the prepared doughs (ca. 450 g) were placed in pans and subjected to 24 h of fermentation with a single puncture. The fermentation was performed at a temperature of 30 °C and relative air humidity of 85% in a fermentation chamber (IBIS, Szubin, Poland). After fermentation was complete, the doughs were baked for 35 min at 240 °C in a baking oven with water vapor (IBIS, Szubin, Poland). In stage III of the study, the baked and cooled breads as well as the remaining collected samples of flour and sourdoughs were dried, ground, and prepared for analyses. The study design scheme is shown in Figure 2.

### 3.3. Carbohydrates Content Determination

The samples were determined for the contents of individual complex carbohydrates: arabinoxylans with the spectrophotometric method, as well as fructans [43] and β-glucans [44] with the enzymatic-spectrophotometric method. For the soluble arabinoxylans assay, aqueous suspensions were extracted for 3 h and, after centrifugation, 1 ml of the liquid fraction was taken for future examination. The method of determination of the content of total, soluble, and insoluble arabinoxylans included boiling the samples for 1 h with sodium chloride, hydrochloric acid, and xylene, and then cooling and separating the solutions. Subsequently, the solutions were mixed with ethyl aniline and ethanol, and then their extinctions were measured at a wavelength of 540 nm and compared with the xylose standard curve. The insoluble arabinoxylans fraction content was calculated as the difference between the total and soluble fractions. The fructan content determination was based on the determination of the fructose content in the samples resulting from the enzymatic breakdown of the fructans. Using a spectrophotometer (GENESYS 10S, Thermo Fisher Scientific, Waltham, MA, USA), the fructose content was measured at a wavelength of λ = 410 nm. The total β-glucan content was determined with the mixed linkage β-glucan assay kit (Megazyme International, Bray, Ireland) following the ICC Standard Method No. 166 [44].

The bread samples were additionally analyzed for starch content with the polarimetric method and for dietary fiber content (including the water-soluble and water-insoluble fractions) with the enzymatic–gravimetric AOAC method [45]. All determinations were performed in duplicate.

### 3.4. Statistical Analysis

The statistical analysis of the results was performed using Statistica 13.3 software (StatSoft, Tulsa, OK, USA). The results are presented as the mean values of two replications. Multiple comparisons were made using the analysis of variance (ANOVA) at p = 0.95. The significance of the differences between the mean values was determined using the Duncan test.

## 4. Conclusions

The awareness of consumers and their interest in the quality of food and nutrition, especially in various ailments of the gastrointestinal tract such as IBS, have immensely increased in the last decade. The present study demonstrated that the treatments applied contributed to an increased total content of arabinoxylans in the breads, and that the inoculation of sourdoughs with potential probiotic strains improved their solubility in water. Bakery products made using sourdough degrade and convert FODMAPs in the rye flour; however, the extent of the degradation of the FODMAPs depends on the fermenting microorganisms. The use of the *S.boulardii* strain may seem prospective as it allowed for the greatest reduction in fructans in the bread in the present study. The high-fiber and low-FODMAP bread prevents bifidobacteria depletion in the intestines and may alleviate IBS symptoms [7]. LAB, despite the ability to ferment sugars into lactic acid, are able to produce proteolytic enzymes. In the bakery industry, LAB enzymes cause gluten protein degradation, resulting in changes of dough rheology and improvement of dough raising capacity, bread volume, and texture. The resulting sourdough bread can be characterized by an extended shelf life as well as improved aroma and taste [46]. Rye bread is an attractive source of dietary fiber as well as other nutrients and bioactive compounds that elicit beneficial nutritional effects because it is consumed in almost all world countries and is available in different variants. Studying the metabolism of carbohydrates by selected strains of microorganisms will allow the future development of innovative cereal products dedicated to consumers with diverse nutritional needs. The current research may contribute to the development of food technology by enabling the use of potential probiotic microorganisms and their metabolic abilities to produce food products with designed composition and quality characteristics.

## Figures and Tables

**Figure 1 molecules-28-01910-f001:**
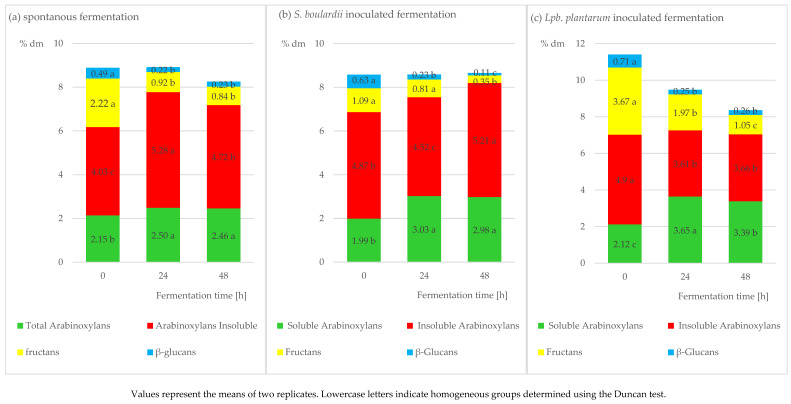
The content of dietary fiber components in the sourdough during fermentation [%]: (**a**) spontaneous fermentation; (**b**) *S. boulardii* inoculated fermentation; (**c**) *Lpb. plantarum* inoculated fermentation; (**d**) *L. rhamnosus* inoculated fermentation; (**e**) *B. coagulans* inoculated fermentation.

**Figure 2 molecules-28-01910-f002:**
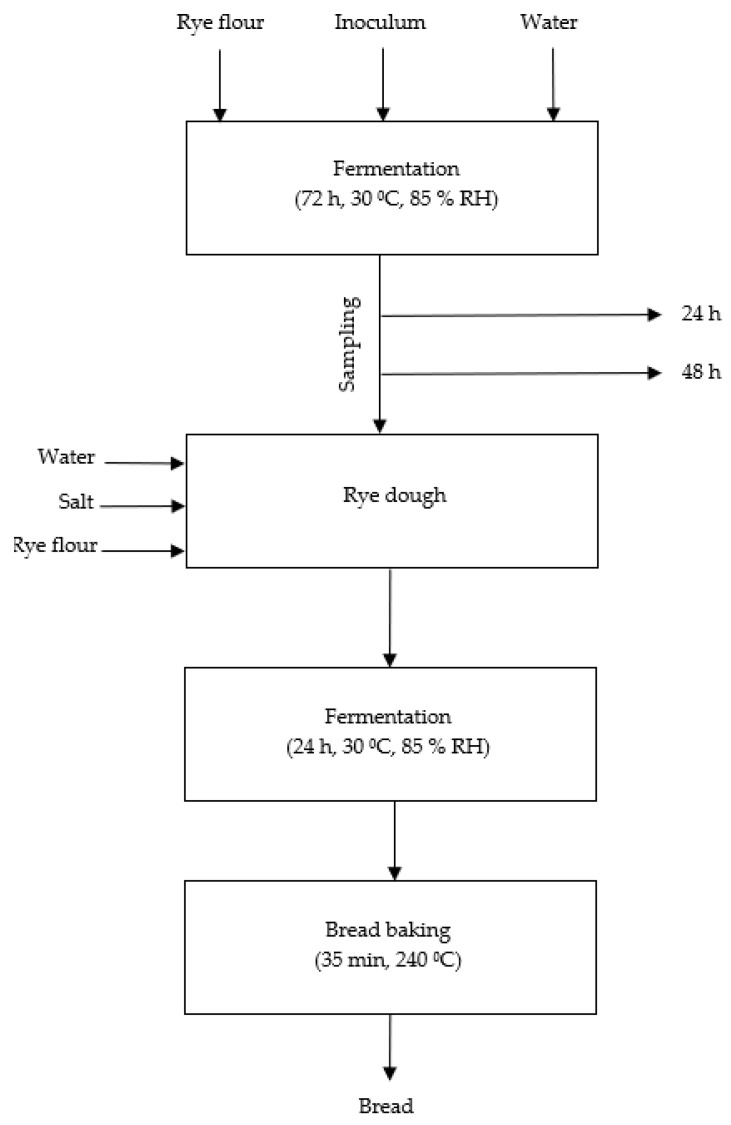
The scheme of the study design.

**Table 1 molecules-28-01910-t001:** The contents of individual polysaccharides and dietary fiber in the dry matter of the rye flour and sourdough breads [% dm].

Type of Fermentation	Starch	Total Arabinoxylans	Soluble Arabinoxylans	Arabinoxylans Insoluble	Fructans	β-Glucans	Total Fiber	Soluble Fiber	Insoluble Fiber
Rye flour	58.67 ± 0.08	8.12 ± 0.05	2.11 ± 0.19	6.01 ± 0.14	3.82 ± 0.012	1.07 ± 0.02	10.39 ± 0.32	1.12 ± 0.40	9.27 ± 0.07
Control bread	57.56 ± 0.91 bc	8.61 ± 0.19 cd	1.42 ± 0.08 c	7.19 ± 0.11 c	1.46 ± 0.19 a	0.30 ± 0.01 e	10.84 ± 0.33 b	1.66 ± 0.50 ab	9.18 ± 0.17 bc
Spontaneous fermentation	57.15 ± 0.33 c	8.39 ± 0.18 d	2.30 ± 0.68 a	6.10 ± 0.86 e	1.32 ± 0.06 a	0.41 ± 0.08 d	9.78 ± 0.92 c	0.63 ± 1.09 c	9.14 ± 0.17 c
*S.boulardii*	58.79 ± 0.11 a	9.65 ± 0.24 b	1.77± 0.03 b	7.89 ± 0.27 b	0.62 ± 0.12 b	0.39 ± 0.01 d	10.01 ± 0.00 c	0.80 ± 0.24 bc	9.21 ± 0.23 bc
*Lpb. plantarum*	58.27 ± 1.27 ab	9.02 ± 0.83 c	2.22 ± 0.39 a	6.81 ± 1.22 cd	1.31 ± 0.62 a	0.49 ± 0.01 b	12.16 ± 0.42 a	2.21 ± 0.17 a	9.95 ± 0.25 a
*L. rhamnosus*	57.61 ± 0.56 bc	8.19 ± 0.11 d	1.90 ± 0.07 b	6.28 ± 0.18 de	1.15 ± 0.12 a	0.53 ± 0.01 a	10.82 ± 0.52 b	1.13 ± 0.35 bc	9.69 ± 0.17 ab
*B.coagulans*	55.91 ± 0.69 d	10.54 ± 0.39 a	1.91 ± 0.25 b	8.60 ± 0.65 a	1.33 ± 0.35 a	0.45 ± 0.06 c	10.26 ± 1.74 bc	1.16 ± 0.95 bc	9.10 ± 0.79 c

Values represent the means of two replicates ± standard deviation. Lowercase letters indicate homogeneous groups determined using the Duncan test.

## Data Availability

Results will be available at corresponding author.

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
