# Peer review of "Rye Dietary Fiber Components upon the Influence of Fermentation Inoculated with Probiotic Microorganisms"

_molecules, 2023, doi:10.3390/molecules28041910_

Round 1

Reviewer 1 Report

1. The paper should be formatted following journal guidelines, including insert Tables and Figures along the text.

2. The linguistic revision of whole manuscript should be carried out.

3. Generally the paper should be  better organized, in order that the different part can be well distributed.

4. The Introduction should be summarized and the novelty character of paper should be marked.

5. The authors should describe the novelty character of this paper respect to other ones present in literature.

6. The authors should decribe better results on rye flour composition  and discussed them in the text respect to previous data present in literature.

7. Results in Figure 1 should be better described in the text.

8. Check decimal points in Table 1

9.  Materials and Methods should be organized in subparagraphs and a major description and details of each methodology should be given, including a graphical scheme of study approach

Author Response

Thank you for your valuable time, careful reading of the manuscript and valuable guidance given in review. The content of the manuscript has been greatly improved.

  1. The paper should be formatted following journal guidelines, including insert Tables and Figures along the text.- The work has been formatted and tables and figures have been placed in the text.
  2. The linguistic revision of whole manuscript should be carried out.- Language corrections have been made.
  3. Generally the paper should be  better organized, in order that the different part can be well distributed.- The paper has been reorganized and subsections have been added.
  4. The Introduction should be summarized and the novelty character of paper should be marked.- A summary of the introduction has been added, highlighting the novelty of the research.
  5. The authors should describe the novelty character of this paper respect to other ones present in literature.- missing fragments have been added.
  6. The authors should decribe better results on rye flour composition  and discussed them in the text respect to previous data present in literature.- an extension of the results and discussion to the parameters of flour have been added.
  7. Results in Figure 1 should be better described in the text.- The description of the results from the figure has been supplemented.
  8. Check decimal points in Table 1- Corrected.
  9. Materials and Methods should be organized in subparagraphs and a major description and details of each methodology should be given, including a graphical scheme of study approach - Subsections in Materials and Methods section have been added. The principles of methodology of used methods and the scheme have also been added.

Thank you again for your valuable comments.

Reviewer 2 Report

This manuscript is a short piece of work, that describes the interest of adding potencial probiotic microorganisms for rye sourdough fermentation and analyses resulting bread dietery fiber evolution. The results reveal that fermentation with the selected microorganisms leads to an increased arabinoxylans content during fermentation and in breads. The authors conclude that S.boulardii strain has the best interest giving its capacity to reduce fructan level in bread.

The manuscript is well written but some issues should be considered before publication.

Introduction:

The criteria for benefiting from the probiotic designation are numerous and the specific strains used in this study are not described or reported to lead to beneficial health effect for the host after ingestion. Therefore, they can not be qualified as probiotics but as “potetial probiotics strains”. Modify along the text.

line 37: "highly capable of binding water". Please modify as insoluble fibers have the lower water-binding capacity among fibers.

Results and discussion

lines 212-220: It can not be concluded from the present data that starch decrease resulted from lower direct acidification. Has fermentation effect on pH evolution or organic acid release been measured ? And has bacterial growth or activity been checked ?

lines 221-233: Dry matter content for both products should be added in order to compare rye flour and cooked breads contents.

Discussion relies heavily on the health effects related to the fibers present in sourdough however this aspect is not investigated in the study. In my opinion, that point should be reduced and have less emphasis. This will allow to get more emphasis on the novelty of this study. The results are not put into perspective with data from literature and the advantage of inoculating probiotics to make sourdough bread should be highlighted.

lines 185, 202, 227, 255, 283, 285 : " L. rhamnosus" check font size

Figure 1 should present standard-error bars and significance letter as subscripts. Correct the legend that indicates a total arabinoxylans content inferior to insoluble arabinoxylans. I think you would say soluble arabinoxylans. In order to help the presentation of results, the layout should be modified to combine all graphs from figure 1 in one page only.

A combination of table 1 and 2 would also help the presentation of results.

3. Materiel and Methods:

This part is reduced to minimum information. It should be organized in sub-part or a scheme should be added for clarity.

Provide more elements on the bacterial strains used to allow their identification. Add details on the form of the micro-organism and details on inoculation preparation if any.

Author Response

Thank you for your valuable time, careful reading of the manuscript and valuable guidance given in review. The content of the manuscript has been greatly improved.

Introduction:

The criteria for benefiting from the probiotic designation are numerous and the specific strains used in this study are not described or reported to lead to beneficial health effect for the host after ingestion. Therefore, they can not be qualified as probiotics but as “potetial probiotics strains”. Modify along the text.- Modified throughout the text.

line 37: "highly capable of binding water". Please modify as insoluble fibers have the lower water-binding capacity among fibers.- This part has been corrected.

Results and discussion

lines 212-220: It can not be concluded from the present data that starch decrease resulted from lower direct acidification. Has fermentation effect on pH evolution or organic acid release been measured ? And has bacterial growth or activity been checked ?- The information about the effect of fermentation on acidity and pH has been added. The pH of sourdoughs was not measured in this study because previous results have found that both inoculated and spontaneously fermenting sourdoughs reached a sufficiently low pH (3.5-4.0) after only 24 hours of fermentation, while extending this time resulted in a relatively constant pH. Bacterial growth was also observed in earlier preliminary studies. On the basis of microscopic observations, the amount of inoculum was determined, that would ensure that the sourdough environment was dominated by the added microorganisms compared to spontaneously fermenting sourdoughs.

lines 221-233: Dry matter content for both products should be added in order to compare rye flour and cooked breads contents.-  All determinations were performed on dried samples and results are presented on a dry weight basis. It has been also specified in the text.

Discussion relies heavily on the health effects related to the fibers present in sourdough however this aspect is not investigated in the study. In my opinion, that point should be reduced and have less emphasis. This will allow to get more emphasis on the novelty of this study. The results are not put into perspective with data from literature and the advantage of inoculating probiotics to make sourdough bread should be highlighted.- The discussion of results has been improved as suggested.

lines 185, 202, 227, 255, 283, 285 : " L. rhamnosus" check font size – Corrected.

Figure 1 should present standard-error bars and significance letter as subscripts. Correct the legend that indicates a total arabinoxylans content inferior to insoluble arabinoxylans. I think you would say soluble arabinoxylans. In order to help the presentation of results, the layout should be modified to combine all graphs from figure 1 in one page only.- Figure 1 has been corrected as suggested. Standard error bars for the content of all components and the reduction of letters responsible for homogeneous groups made the graphs less readable. The charts do not fit legibly on one page, they are spread over two.

A combination of table 1 and 2 would also help the presentation of results.- Tables 1 and 2 have been compiled into one.

Materiel and Methods:

This part is reduced to minimum information. It should be organized in sub-part or a scheme should be added for clarity.- The Materials and Methods section has been reorganized and supplemented with subsections, scheme and description of carbohydrates determination principles.

Provide more elements on the bacterial strains used to allow their identification. Add details on the form of the micro-organism and details on inoculation preparation if any.- Information about microorganisms and their addition has been added.

Thank you again for your valuable comments.

Reviewer 3 Report

The author’s investigation on rye dietary fiber components influence of fermentation with probiotic described significant results and the future potentials. The manuscript is well-written; the findings are well-explained with relevant references. I have some suggestions for enhancing the manuscript readership.

  • Materials and methods need elaborations.
  • Figure 1 is not appropriately presented. There are no unit labels on the graph axes. The color scheme also lacks distinction.
  • Conclusion needs to be improved keeping in mind the results of the study and future perspectives.

Author Response

Thank you for your valuable time, careful reading of the manuscript and valuable guidance given in review. The content of the manuscript has been greatly improved.

Materials and methods need elaborations.-  The Materials and Methods section has been reorganized and supplemented with subsections, scheme and description of carbohydrates determination principles.

Figure 1 is not appropriately presented. There are no unit labels on the graph axes. The color scheme also lacks distinction.- The Figure has been corrected as suggested.

Conclusion needs to be improved keeping in mind the results of the study and future perspectives.- Conclusions have been supplemented and corrected.

Thank you again for your valuable comments.

Round 2

Reviewer 1 Report

The paper is suitable for publication

Author Response

Thank you for your valuable comments.

Reviewer 2 Report

The manuscript has been notably improved.

Author Response

Thank you for your valuable comments.